

# Assessing the effect of cardiovascular disease on work productivity and financial loss among school teachers in Peninsular Malaysia: a nested case-control study

Jun Fai Yap[1], Foong Ming Moy[1], Wan Azman Wan Ahmad[2] and Yin Cheng Lim[1]

[1] Department of Social and Preventive Medicine, Faculty of Medicine, Universiti Malaya, Kuala Lumpur, Malaysia
[2] Cardiology Unit, Department of Medicine, Faculty of Medicine, Universiti Malaya, Kuala Lumpur, Malaysia

Corresponding author
Yin Cheng Lim, limyc@ummc.edu.my

## ABSTRACT

**Background**. School teachers may have an increased risk of cardiovascular disease (CVD), potentially affecting their work productivity. However, limited data exists on the impact of CVD on teachers' productivity in Malaysia. Our objectives were to assess work productivity loss (absenteeism and presenteeism) as well as to determine the associated annual monetary loss among school teachers who experienced incident CVD in Peninsular Malaysia.

**Methods**. We adopted a nested case-control design within a cohort of school teachers. Working teachers from six states of Peninsular Malaysia, and had experienced incident CVD before a right-censored date (31st December 2021) were defined as cases. Incident CVD was operationally defined as the development of non-fatal acute coronary syndrome (ACS), stroke, congestive cardiac failure, deep vein thrombosis or peripheral arterial disease before the censored date. Controls were working teachers who did not acquire an incident CVD before the similar right-censored date. All controls were randomly selected, with a ratio of one case to four controls, from among the working teachers in one of the states in Peninsular Malaysia. We used a shortened version of the Malay-validated World Health Organization-Health and Work Performance Questionnaire (WHO-HPQ) to estimate the workplace productivity effect among teachers with incident CVD (cases). The same questionnaire was distributed to teachers in a single state of Peninsular Malaysia who did not experience incident CVD (controls). Absenteeism, presenteeism and annual monetary loss were computed based on the scoring rules in the WHO-HPQ. Analysis of covariance was performed with covariate adjustment using propensity scores. The bootstrapping method was applied to obtain better estimates of marginal mean differences, along with standard errors (SE) and appropriate effect sizes.

**Results**. We recruited 48 cases (baseline mean age = 42.4 years old, 54.2% females) and 192 randomly selected controls (baseline mean age = 36.2 years old, 99.0% females). The majority of the cases had ACS (73.9%). No significant difference was observed in absenteeism between cases and controls. The mean self-rated job performance score was lower for cases (7.63, SE = 0.21) compared to controls (8.60, SE = 0.10). Marginal mean scores of absolute presenteeism among cases (76.30) were lower ($p < 0.05$, eta squared = 0.075) than controls (85.97). The marginal mean annual cost of presenteeism

was higher in cases (MYR 21,237.52) compared to controls (MYR 12,089.74) ($p < 0.05$, eta squared $= 0.082$).

**Conclusion**. Absolute presenteeism was lower among school teachers who experienced incident CVD and the annual cost of presenteeism was substantial. Implementing supportive work strategies in school settings is recommended to increase absolute presenteeism, which can lead to a reduction in the annual cost of presenteeism among teachers experiencing incident CVD.

# INTRODUCTION

Cardiovascular disease (CVD) is a general term for several medical disorders that affect the heart or blood vessels. It remains a global health threat affecting individuals of all races and selected occupations (*Havranek et al., 2015*). School teachers are not exempted from acquiring CVD, possibly attributed to the stressful teaching environment (*Riechmann-Wolf et al., 2021*).

Different types of CVD have negative effects on school teachers' work productivity. For example, teachers with acute coronary syndrome (ACS) may have reduced exercise tolerance to climb stairs in schools or conduct physical education classes. Based on a meta-analysis, nearly 20% of ACS survivors did not resume work (*Kai et al., 2022*). Up to 75% of stroke survivors struggled in controlling their hand dexterity, making it difficult for teachers to write on the board or manipulate classroom materials. Stroke-related dysarthria may also impair an effective verbal communication and further jeopardize the teaching process. Approximately 44% of stroke survivors at working age were unable to return to work, with the consequent productivity loss of nearly 1.9 billion dollar in United Kingdom (UK) (*Saka, McGuire & Wolfe, 2009*). Congestive cardiac failure (CCF) survivors with reduced ejection fraction easily felt fatigued, limiting teachers' ability to perform physically strenuous activities in the classroom. Most teachers who were treated for deep vein thrombosis (DVT) may not achieve full functional recovery post-hospitalization. This speculation is supported by a large population-based cohort study among 66,005 working adults, which found that DVT patients had an 80% higher risk of work-related disability compared to non-DVT individuals (*Braekkan et al., 2016*). Teachers with peripheral arterial disease (PAD) may not be able to walk continuously due to claudication pain. Thus, they may not be physically fit to lead field trips or extracurricular activities that require prolonged walking.

Absenteeism is a regularly used indicator to assess work productivity, which is defined as not remaining at work as scheduled. Notably the absenteeism rates of school teachers ranged from 15% to 45% in countries of Africa (*O'Sullivan, 2022*). In particular, teachers who self-reported poor health had a 70% higher odds of absenteeism (*Coledam et al., 2021*).

In contrast, presenteeism or working-while-ill is characterized by sick workers turning up for work but not functioning fully at the workplace due to an underlying disease (*Johns, 2010*). The prevalence of sickness presenteeism among school teachers is generally higher at 57.1% to 65.2% (*Dudenhöffer et al., 2017*; *Rojas-Roque & López-Bonilla, 2022*). Presenteeism seems to be common in the teaching profession (as compared to other professions which do not handle people) as it is challenging to assign a substitute to teach without knowing the study progress of students (*Marklund et al., 2021*). By addressing workplace productivity loss among the affected teachers, a healthier learning environment can be fostered in the school community, leading indirectly to better students' academic performance.

On average, UK workers spend 35 hours weekly to work but lost only 30 days to presenteeism and absenteeism in a year (*Wee et al., 2019*). Malaysian workers spend about 44 hours weekly to work but 66 days are lost annually to presenteeism and absenteeism. This workplace productivity loss translated to about 4.53% of Malaysia's gross domestic product in 2015 (*Rasmussen, Sweeny & Sheehan, 2016*). Plausible explanations of the comparatively lower workplace productivity loss in UK include existence of comprehensive workplace wellness programmes in most regions of UK and better work culture engaging workers' participation, leading to generally better workers' mental or physical wellbeing (*Hassan et al., 2009*). In Malaysia, workplace wellness programme is more prevalent in larger organizations with more financial resources and may not be present in small and medium-sized enterprises (*Mohamed et al., 2022*; *Moy, Sallam & Wong, 2006*).

The California Teachers Study (*Pacheco et al., 2020*; *Willey et al., 2017*), a large prospective teacher cohort in the United States (US) only focus on establishing the causal relationship spatiotemporally between lifestyle factors and incident CVD. There is no detailed description of CVD impact on teachers' work productivity in schools across the Europe, US or the Asia continents. Thus, the effect of CVD on workplace productivity remains unclear among school teachers. This study aims to determine the workplace effect of CVD and its monetary implication among school teachers in Peninsular Malaysia. By estimating the economic consequences of incident CVD among the school teachers, proposal of workplace wellness programme and improvement of current sick leave policy in Malaysian schools can be justified to reduce or prevent workplace productivity loss.

## MATERIALS & METHODS

### Study design
Nested case-control study design was used to estimate the workplace productivity effect (absenteeism and presenteeism) as well as its annual costs among school teachers with incident CVD across six states (Selangor, Kuala Lumpur, Malacca, Terengganu, Penang and Johor) of Peninsular Malaysia.

### Data source
Baseline data was extracted from a cohort of school teachers of the 'Clustering of Lifestyle Risk Factors and Understanding its Association with Stress on Health and Wellbeing among Malaysian School Teachers' (CLUSTer) study (*Moy et al., 2014*). These teachers

were followed up from their recruiting date (2013–2014) until a right-censored date (31st December 2021) to detect non-fatal incident CVD. Basic sociodemographic details and baseline medical conditions were collected during the recruitment period. The included baseline sociodemographic characteristics were age, states, marital status, gender, ethnicity (Malay, Chinese, Indian, others) and handphone numbers. Medical conditions of interest including presence of laboratory-confirmed diabetes mellitus (DM), self-reported hypertension, high low-density lipoprotein cholesterol (LDL-C) and high triglyceride were reported. Teachers who had self-reported CVD at baseline were excluded, potentially minimizing reverse causality due to improved health behaviour following an incident CVD.

Incident CVD was operationally defined as the development of non-fatal ACS (including unstable angina, non ST-segment elevation myocardial infarction and ST-segment elevation myocardial infarction), stroke (including ischemic subtype, hemorrhagic subtype and transient ischemic attack), CCF, DVT or PAD before 31st December 2021. The incident CVD diagnoses were confirmed *via* data linkage across three national registries, namely the National Stroke Registry (*Aziz et al., 2015*), the National CVD-ACS Database (*Ahmad et al., 2011*) and the National Hospital Admission Database which captured more than 95% of all public hospital admissions' data in Malaysia.

Teachers (non-retired) who had incident CVD before the right-censored date were defined as cases. In contrast, controls were working teachers who did not acquire an incident CVD before the similar right-censored date. All controls were randomly selected (in one case to four controls ratio) from among the working teachers in one of the Peninsular Malaysia's states (Johor). Unforeseen logistical and financial issues resulted in only teachers from one state (Johor) completing the distributed World Health Organization-Health and Work Performance Questionnaire (WHO-HPQ). Nevertheless, controls from Johor state alone were deemed comparable with other states. The case-to-control ratio was fixed at 1:4 since this would increase the power of the study and little statistical power would be gained beyond four controls.

The sample size calculation was performed using the OpenEpi statistical calculator. Based on a Swiss workforce survey, with a mean difference of −0.075 in health-related productivity loss, power of 80% and at a 95% confidence interval (CI), a minimum of 42 cases and 172 controls (after inflation of 10% attrition rate) were required (*Brunner et al., 2019*).

## Questionnaire

All questions were typed on the Research Electronic Data Capture (REDCap) platform and emailed to eligible teachers (including both cases and controls) *via* the school principals. REDCap is a secured web-based application in managing online surveys and collecting survey data (*Patridge & Bardyn, 2018*). All teachers who did not respond to the emails after a week were phone-interviewed by using the same set of WHO-HPQ, since telephone-administration mode of health-related questionnaire had been proven to be as valid as the self-administered mode (*Lungenhausen et al., 2007*). The assessment, conducted using

**Table 1  Short version of World Health Organization-Health and Work Performance Questionnaire.**

|  | Question 1 | Question 2 | Question 3 | Question 4 | Question 5 | Question 6 | Question 7 | Question 8 |
|---|---|---|---|---|---|---|---|---|
| **Parameters** | Monthly salary | Presence of new chronic disease | Presence of new acute disease | Expected work (hours/ week) | Days of missed work | Actual work (hours/ month) | Others' job performance during the past one month | Self-rated job performance during the past one month |
| **Involved in calculation of** | Annual cost of absenteeism Annual cost of presenteeism | – | – | Absolute absenteeism Relative absenteeism | – | Absolute absenteeism Relative absenteeism | Relative presenteeism | Absolute presenteeism |

the WHO-HPQ, occurred between October 2022 and December 2022. All data gathered through the phone interview session were also inserted into REDCap by the interviewer.

## Outcome

The locally validated, short Malay version of WHO-HPQ was used to measure workplace productivity among our teachers (*Kessler et al., 2004*; *Muhamad Hasani & Chinna, 2015*). It has good internal consistency with Cronbach's alpha of 0.78 and mean inter-item correlation of 0.53 (*Muhamad Hasani & Chinna, 2015*). The questionnaire consisted of eight questions (in the Malay language) related to absenteeism and presenteeism (see Appendix).

Absenteeism is divided into absolute and relative absenteeism. For absolute absenteeism calculation of an individual (expressed in hours per month), value of Question 4 (expected work per week) is multiplied by four before deducting value of Question 6 (actual work per month) (absolute absenteeism = 4 × Question 4–Question 6) (Table 1). If the final score is a negative number, it means the teacher had worked more than expected. Otherwise, any positive number represents the number of hours the teacher needed to work. Relative absenteeism is a benchmark or reference point to offer a more comparative analysis of absence. Its formula expresses the expected hours to work of the same individual in a ratio (Relative absenteeism = (4 × Question 4–Question 6)/ (4 × Question 4)). The result is expected to range from negative values (worked beyond expectation) to 1.0 (always absent). Relative hours of work is equivalent to one minus relative absenteeism. (Relative hours of work = (1–Relative absenteeism)).

Presenteeism is also subcategorized into absolute and relative presenteeism. Absolute presenteeism value is obtained by multiplying Question 8 (self-rated job performance) with 10 (Absolute presenteeism = 10 × Question 8). The minimum score of absolute presenteeism is 0, indicating a total lack of productivity during work. In contrast, the maximum score is 100, in which there is no lack of productivity during work. For relative presenteeism, the actual job performance of oneself (Question 8) is divided by the job performance of most workers at the same job (Question 7) (Relative presenteeism = Question 8/Question 7). If Question 7 is graded as zero, an automatic result of 2.0 is assigned as division by zero yields an infinite number. A result of 2.0 is designated as the

upper bound, signifying the best relative productivity of a worker (equivalent to 200% or more of other workers' job performance) (*Suzuki et al., 2014*). A result of 0.25 is taken as the lower bound (25% of less of other workers' job performance). If both Question 7 and 8 are graded as zero, then the relative presenteeism of 1.0 is assigned since division of zero by zero is impossible mathematically.

Estimated monetary loss of workplace productivity is calculated for a year. The formulas of annual cost secondary to absenteeism and presenteeism are listed below:

- Annual cost of absenteeism (in MYR) = Current monthly salary × (Days of work missed in the past one month/28) × 12
- Annual cost of presenteeism (in MYR) = Current monthly salary × [(10–Self-rated work performance in the past one month)/10] × 12

Days of work missed in the past one month is derived from Question 5 whereas self-rated job performance in the past one month is obtained from Question 8.

## Statistical analysis

Bootstrapping was used to test the hypothesis if there were any marginal mean differences for both cases and controls. Bias-corrected accelerated (Bca) 95% CI for the marginal mean differences was generated from 5,000 bootstrap samples. The number of 5,000 was chosen as it was considered large enough to yield stable estimates (*Feifel & Dobler, 2021*).

Propensity score was used to adjust for the confounding effects of sociodemographic characteristics (age, marital status, gender, ethnicity) and other medical conditions (presence of laboratory-confirmed DM, self-reported hypertension, high LDL-C, high triglyceride) when estimating the impact of incident CVD on workplace productivity. A Cox regression model was fitted with incident CVD as the dependent variable and the selected confounders (marital status, gender, ethnicity, presence of laboratory-confirmed DM, self-reported hypertension, high LDL-C, high triglyceride) as independent variables to derive the predicted probabilities. These predicted probabilities of the fitted regression model served as estimated propensity scores. Analysis of covariance (ANCOVA) was then used to estimate marginal mean differences with standard errors (SE) and effect sizes (eta squared) by including propensity scores as a new covariate. Based on Cohen's criteria, eta-squared values of 0.01, 0.06 and 0.14 indicated small, medium and large effect sizes respectively (*Burdick-Will & Logan, 2017*).

Complete case analysis was used to handle the missing data. All data analysis was performed using Statistical Package for the Social Sciences (SPSS, version 20). *P*-values were based on two-sided tests and $p < 0.05$ was taken as statistically significant. Ethical approval was obtained from the National Medical Research Registry, Medical Research and Ethics Committee, Ministry of Health Malaysia (Reference number: NMRR ID-22-00811-IYL) and the Ministry of Education Malaysia (KPM.600-3/2/3-eras (13109)).

## RESULTS

From the data linkage with three registries as mentioned above, there were 209 incident CVD cases. Eleven (5.3%) of them had passed away. Only 72 (34.4%) responded to our

**Table 2** Baseline characteristics of cases (with incident cardiovascular disease) and controls (without incident cardiovascular disease).

|  | Cases ($n = 48$) | Controls ($n = 192$) |
|---|---|---|
| **Age (Mean ± Standard deviation)** | 42.4 ± 6.5 | 36.2 ± 6.9 |
| **States ($n = 240$)** | **Frequency (Percentage)** | **Frequency (Percentage)** |
| Johor | 16 (33.3) | 192 (100) |
| Kuala Lumpur | 2 (4.2) | 0 |
| Malacca | 5 (10.4) | 0 |
| Penang | 5 (10.4) | 0 |
| Selangor | 13 (27.1) | 0 |
| Terengganu | 7 (14.6) | 0 |
| **Marital status ($n = 240$)** | | |
| Single | 5 (10.4) | 29 (15.1) |
| Married | 42 (87.5) | 161 (83.9) |
| Divorced or widow | 1 (2.1) | 2 (1.0) |
| **Gender ($n = 240$)** | | |
| Females | 26 (54.2) | 190 (99.0) |
| Males | 22 (45.8) | 2 (1.0) |
| **Ethnicity ($n = 240$)** | | |
| Chinese | 4 (8.3) | 22 (11.5) |
| Malay | 39 (81.2) | 161 (83.9) |
| Indians and others | 5 (10.4) | 9 (4.7) |
| **Laboratory-confirmed diabetes mellitus ($n = 224$)** | | |
| No | 37 (78.7) | 172 (97.2) |
| Yes | 10 (21.3) | 5 (2.8) |
| **Self-reported hypertension ($n = 226$)** | | |
| No | 40 (83.3) | 173 (97.2) |
| Yes | 8 (16.7) | 5 (2.8) |
| **High low-density lipoprotein cholesterol ($n = 212$)** | | |
| No | 9 (19.1) | 70 (42.4) |
| Yes | 38 (80.9) | 95 (57.6) |
| **High triglyceride ($n = 224$)** | | |
| No | 27 (57.4) | 163 (92.1) |
| Yes | 20 (42.6) | 14 (7.9) |

telephone calls and completed the questionnaire. There were 25 (12.0%) responded to our calls but declined to answer the questionnaire. The rest of the incident CVD cases were not contactable either did not answer to the calls (29.2%), phone number not in service (14.8%) or phone number not available (4.3%).

From the 72 teachers who answered the questionnaires, only 48 (66.7%) were still working. Slightly more than half of the cases were females (54.2%) with mean (±standard deviation) baseline age of 42.4 (±6.5) years old (Table 2). Ninety-nine percent of the controls were females, with a mean (±standard deviation) age being 36.2 (±6.9) years old.

**Table 3  Absenteeism and presenteeism among working school teachers.**

| | Cases (n = 48) | | Controls (n = 192) | | Bootstrapped approach | | | |
|---|---|---|---|---|---|---|---|---|
| | Marginal mean (Standard error) | [a]95% CI | Marginal mean (Standard error) | [a]95% CI | Mean difference (Standard error) | [*]Bias-corrected accelerated 95% CI | [+]p value | [b]Eta squared (Effect size) |
| Expected work (hours/week) | 35.35 (6.56) | 22.42, 48.29 | 40.00 (3.22) | 33.65, 46.34 | 4.65 (0.29) | −7.05, 20.13 | 0.53 | 0.008 (Small) |
| Actual work (hours/ month) | 125.66 (7.83) | 110.23, 141.10 | 137.24 (3.84) | 129.67, 144.81 | 11.58 (9.84) | −6.88, 31.48 | 0.22 | 0.010 (Small) |
| Others' job performance during the past one month | 7.94 (0.23) | 7.49, 8.39 | 8.37 (0.11) | 8.15, 8.59 | 0.43 (0.32) | −0.15, 1.10 | 0.18 | 0.013 (Small) |
| Self-rated job performance during the past one month | 7.63 (0.21) | 7.21, 8.05 | 8.60 (0.10) | 8.39, 8.80 | 0.97 (0.29) | 0.39, 1.55 | **0.002** | 0.075 (Medium) |
| Absolute absenteeism | 15.74 (26.99) | −37.47, 68.95 | 22.76 (13.24) | −3.35, 48.86 | 7.01 (30.23) | −42.77, 73.77 | 0.83 | 0.004 (Small) |
| Relative absenteeism | −0.12 (0.34) | −0.78, 0.55 | −0.55 (0.17) | −0.87, −0.22 | −0.43 (0.29) | −0.93, 0.14 | 0.14 | 0.008 (Small) |
| Relative hours of work | 1.12 (0.34) | 0.45, 1.78 | 1.55 (0.17) | 1.22, 1.87 | 0.43 (0.29) | −0.24, 0.99 | 0.14 | 0.008 (Small) |
| Absolute presenteeism | 76.30 (2.12) | 72.13, 80.47 | 85.97 (1.04) | 83.92, 88.01 | 9.67 (2.94) | 4.06, 15.56 | **0.001** | 0.075 (Medium) |
| Relative presenteeism | 1.03 (0.04) | 0.95, 1.10 | 1.04 (0.02) | 1.00, 1.07 | 0.01 (0.07) | −0.17, 0.15 | 0.90 | 0.006 (Small) |

**Notes.**
[a]Bonferroni method was used to adjust and construct the 95% CI.
[*]Bias-corrected accelerated 95% CI was obtained after running 5,000 bootstrapped samples.
[+]p value was reported based on ANCOVA including propensity scores as a covariate.
[b]Benchmarks of small (Eta squared = 0.01), medium (Eta squared = 0.06) and large (Eta squared = 0.14) effect sizes were defined by Cohen.
Bold values denote statistical significance at the $p < 0.05$ level.

Majority of cases (81.2%) and controls (83.9%) were of Malay ethnicity. Most cases and controls were predominantly married. Cases had higher proportions of laboratory-confirmed DM (21.3%), self-reported hypertension (16.7%), high LDL-C (80.9%) and high triglyceride (42.6%), as compared to controls.

Majority of the cases had ACS (73.9%), followed by stroke (15.2%), CCF (4.3%), DVT (4.3%) and PAD (2.2%). Table 3 demonstrates workplace productivity among these working teachers. The expected means duration of work in a week by the cases and controls were 35.35 (SE = 6.56) hours and 40.00 (SE = 3.22) hours, respectively. The expected means duration of work in a month by the cases and controls were 141.4 hours and 160 hours, respectively. However, the actual self-reported durations of work in a month by the cases and controls were lower at 125.66 (SE = 7.83) hours and 137.24 (SE = 3.84) hours, respectively. No statistically significant differences were noted between cases and controls for such parameters.

Mean scores of absolute absenteeism for cases and controls were valued at 15.74 (SE = 26.99) and 22.76 (SE = 13.24) respectively. In contrast, mean relative absenteeism for

**Table 4  Estimated indirect monetary loss of absenteeism and presenteeism among working school teachers.**

| | Cases (n = 48) | | Controls (n = 192) | | Bootstrapped approach | | | |
|---|---|---|---|---|---|---|---|---|
| | Marginal mean (Standard error) | [a]95% CI | Marginal mean (Standard error) | [a]95% CI | Mean difference (Standard error) | [*] Bias-corrected accelerated 95% CI | [+] p value | [b]Eta squared (Effect size) |
| Annual cost of absenteeism (MYR) | 4,249.55 (4,062.72) | −3,759.62, 12,258.72 | 5,882.80 (1,992.89) | −1,954.06, 9,811.55 | 1,633.26 (3,551.05) | −5,566.61, 8,488.12 | 0.66 | 0.001 (Small) |
| Annual cost of presenteeism (MYR) | 21,237.52 (1,824.24) | 17,641.26, 24,833.79 | 12,089.74 (894.84) | 10,325.66, 13,853.82 | −9,147.78 (2,590.78) | −14,368.74, −4,134.99 | **0.001** | 0.082 (Medium) |

**Notes.**

[a]Bonferroni method was used to adjust and construct the 95% CI.

[*]Bias-corrected accelerated 95% CI was obtained after running 5,000 bootstrapped samples.

[+]p value was reported based on ANCOVA including propensity scores as a covariate.

[b]Benchmarks of small (Eta squared = 0.01), medium (Eta squared = 0.06) and large (Eta squared = 0.14) effect sizes were defined by Cohen.

Bold values denote statistical significance at the $p < 0.05$ level.

cases and controls were negatively valued at −0.12 (SE = 0.34) and −0.55 (SE = 0.17), respectively. There was no difference in absenteeism between cases and controls.

The self-rated job performance score during the past one month and the mean scores of absolute presenteeism were significantly different between cases and controls. The self-rated job performance score was lower [7.63 (SE = 0.21)] for cases compared to controls [8.60 (SE = 0.10)]. The mean scores of absolute presenteeism for cases (76.30, SE = 2.12) was also lower than controls (85.97, SE = 1.04) ($p = 0.001$, eta squared = 0.075). However, the mean relative presenteeism for cases and controls had no significant differences. In terms of the estimated monetary loss, the annual cost of presenteeism was different ($p = 0.001$, eta squared = 0.082) between cases (MYR 21,237.52) and controls (MYR 12,089.74) (Table 4).

## DISCUSSION

Overall, there was differential effect of an incident CVD on work productivity of the affected school teachers. Absenteeism portrayed no difference between cases and controls. As compared to controls, cases had lower self-rated job performance, lower absolute presenteeism and higher annual cost of presenteeism. However, there was no mean difference of relative presenteeism between cases and controls, suggesting a situation where teachers were still working despite being unwell, but their productivities were reduced within an acceptable range compared to their usual performance.

Both absolute and relative absenteeism did not differ between the cases or controls. In other words, incident CVD exerted impact onto teachers' work absence irrespective of the diagnosis status. As compared to the average absolute absenteeism score of patients with common mental disorders in India (45.0 hours per month) (*Kondapura et al., 2023*), our teachers with incident CVD had lower scores (15.74 hours per month). As government servants, public school teachers in Malaysia are entitled for 90 days of paid sick leave, which might explain a lower absolute absenteeism among our cases. Another possible explanation was that school teachers with incident CVD were granted similarly adequate medical leave

(as compared to controls who might be absent due to other illnesses) by the treating physicians before being discharged from hospitals. Notably mean relative absenteeism was negatively valued in both cases and controls, signifying school teachers worked more than expected regardless of incident CVD. Such phenomenon might be a norm especially in schools where ratio of teachers to students was disproportionately low (*Burdick-Will & Logan, 2017*).

The concordance result of lower self-rated job performance and lower absolute presenteeism was expected since absolute presenteeism was a multiplier of the self-rated job performance by 10. In general, subjective self-rated wellbeing was associated with a lifetime CVD risk (*Boehm & Kubzansky, 2012*). Although different measuring tool was used, self-rated job performance (influenced by different degrees of job strain or job insecurity) was considered a strong predictor of incident CVD cases among middle-aged working women (*Slopen et al., 2012*). In this context, CVD patients often self-rated themselves to be poorer in own health or job parameters (*Orimoloye et al., 2019*).

Whilst the majority of teachers demonstrate dedication and commitment to their noble educational work, persistently working while unwell can pose health risks. Instances of low absolute presenteeism resulting from incident CVD among affected school teachers could potentially contribute to heightened emotional distress, subsequently impacting their overall job satisfaction. Left unaddressed, this could ultimately lead to burnout stemming from excessive presenteeism. Furthermore, colleagues might be compelled to compensate for the reduced workplace productivity of those who continue to work while unwell, which has the potential to create feelings of resentment and job-related stress. Consequently, the quality of interactions between teachers and students could be adversely affected.

Nevertheless, overestimation in annual cost of presenteeism could not be ruled out in the cases as the reduced job performance values were directly multiplied by the estimated reduced hours equivalent by the monthly income (*Pauly et al., 2008*). This finding was in contrast to the previously published research which concluded a lower monetary value imposed by presenteeism (than absenteeism) among healthcare workers (*Rantanen & Tuominen, 2011*).

The mean absolute presenteeism score among patients with underlying arthritis or similar rheumatological conditions was 74.1 (*AlHeresh et al., 2017*), comparable with our cases with incident CVD. However, another study which investigated employees with mental illness found a much lower mean absolute presenteeism score at 42.5 (*Suzuki et al., 2014*).

Conventionally, WHO-HPQ is recommended to measure health-related workplace productivity among employees suffering from chronic illnesses, but it has not been utilized to assess CVD patients (*Kessler et al., 2004*). A validation study confirmed that absolute presenteeism was the best parameter in WHO-HPQ which correlated well with health indicators (*Scuffham, Vecchio & Whiteford, 2014*). However, relative presenteeism which is a ratio of own job performance to others' job performance may introduce errors as objective assessment of other colleagues' job performance often could not be made, partially explaining the non-significance of relative presenteeism. While relative presenteeism showed no difference between cases and controls, the quality of teaching

among affected teachers might still be compromised leading to impaired learning process among the students.

The annual cost of presenteeism is anticipated to rise with a decrease in absolute presenteeism. This is due to the necessity of subtracting the parameter associated with presenteeism, specifically 'self-rated work performance,' in the formula for calculating the annual cost of presenteeism. In 2017, an estimated US$ 378 billion (MYR 1.76 trillion) was lost due to CVD in the US while the direct healthcare cost of CVD in Malaysia was reported to be approximately MYR 3.93 billion during the same year (*Malaysia MoH, 2022*). No other comparison could be made on presenteeism caused by CVD as there was no detailed cost analysis in this aspect. Thus far, most published studies in US focused on workplace absenteeism and did not analyse on presenteeism (*Song et al., 2015*). Presenteeism may be more difficult to quantify since there was no single measure outcome which was deemed as a gold standard.

While interpreting the findings, there are a few limitations which warrant discussion. Firstly, there was a small event rate with high non-response rate. Therefore, we could not conduct further regression analyses. It was also not possible to completely eliminate selection bias, specifically sampling bias. Matching on age or gender was not done as a matched design reduced effective sample size. Nonetheless, the use of advance statistical methods such as bootstrapping and ANCOVA adjusting for propensity scoring as a covariate reduced the error variance. In addition, random selection of controls minimized selection bias since the controls were chosen without preconceived criteria based on multiple exposures. The short version of WHO-HPQ may not be able to capture the complexity and nuances of health-related work productivity comprehensively. Nevertheless, the short version of WHO-HPQ was more likely to be fully completed by the participants. Finally, the cost estimates of productivity loss were based on recalls and therefore subjected to self-report bias.

This study outlined the importance in mitigating incident CVD and provided convincing evidence for CVD prevention programmes. Goal three for health in Sustainable Development Goals (SDG) emphasized on ensuring healthy lives besides promoting wellbeing at all ages, while one of the clause of goal four in SDG stated that qualified teachers should be increased by 2030. Since a higher absolute presenteeism score denotes a lesser extent of lost performance, strategies to increase absolute presenteeism or decrease annual cost of presenteeism should be designed and implemented for better workplace wellness among the teachers.

## CONCLUSIONS

Absolute presenteeism was significantly different between cases and controls, with absenteeism parameters and relative presenteeism showing no difference. Absolute presenteeism was lower among school teachers who had incident CVD and the annual cost of presenteeism was costlier compared to teachers without incident CVD. Considering the substantial financial losses, it is advisable to implement strategies at increasing absolute presenteeism, which can lead to a reduction in the annual cost of presenteeism among school teachers experiencing incident CVD.

A recommended approach involves expanding workplace wellness programmes in Peninsular Malaysia, with a particular focus on school teachers who exhibit at least one traditional cardiovascular risk factor. These initiatives could include screening them for other common risk factors, accurately diagnosing symptomatic ones and intervening earlier to reduce the risk of acquiring incident CVD. Additionally, an alternative recommendation is to encourage the utilization of sick leave for teachers who have experienced an incident CVD. By allowing affected teachers to take the necessary time off to recover, the occurrence of absolute presenteeism following an incident CVD is expected to increase over time.

## ACKNOWLEDGEMENTS

We express our utmost gratitude to all the teachers who participated in this study and would like to acknowledge the Ministry of Education Malaysia. Our sincere appreciation goes to the Ministry of Health Malaysia institutions, namely the Health Informatics Centre, National Heart Association Malaysia and the Clinical Research Centre at Hospital Sultanah Nur Zahirah, for their close cooperation throughout the data linkage process. We would also like to thank the Director General of Health Malaysia for his permission to publish this article.

### Funding

The authors received no funding for this work.

### Competing Interests

The authors declare there are no competing interests.

### Author Contributions

- Jun Fai Yap conceived and designed the experiments, performed the experiments, analyzed the data, prepared figures and/or tables, authored or reviewed drafts of the article, and approved the final draft.
- Foong Ming Moy performed the experiments, authored or reviewed drafts of the article, and approved the final draft.
- Wan Azman Wan Ahmad performed the experiments, authored or reviewed drafts of the article, and approved the final draft.
- Yin Cheng Lim performed the experiments, authored or reviewed drafts of the article, and approved the final draft.

### Human Ethics

The following information was supplied relating to ethical approvals (i.e., approving body and any reference numbers):

Ethical approval was obtained from the National Medical Research Registry (Reference number: NMRR ID-22-00811-IYL) and the Ministry of Education Malaysia (KPM.600-3/2/3-eras (13109).

## Data Availability

The raw data are in available in the Supplementary File.

## Supplemental Information

Supplemental information for this article can be found online at http://dx.doi.org/10.7717/peerj.16906#supplemental-information.

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
