# Peer review of "Assessing the effect of cardiovascular disease on work productivity and financial loss among school teachers in Peninsular Malaysia: a nested case-control study"

_PeerJ, doi:10.7717/peerj.16906_

## Round 0.1 · original submission · Minor Revisions

If you can please complete the minor revisions and submit within the next two weeks, we will be able to consider it for publication.

Reviewer 1 ·

Basic reporting

Good and references are up to date.

Experimental design

1. The use of terms like “cases” and “controls” is clear in the context.
2. In the methods section, it is necessary to clarify why controls were solely selected from Johor, as this may introduce bias.
3. Providing additional details on the rationale behind the sample size calculation and why a 1:4 case-control ratio was chosen would be helpful.

Validity of the findings

No comment

Additional comments

I would like to congratulate you on your fascinating study and well-written manuscript.

Comments:

1. In the results section, the data is presented well. However, there is some ambiguity in the presentation, particularly in the statement "Nearly 99% of them were females," which contradicts the previously stated 54.2% female demographic. This statement needs clarification to ensure consistency and avoid confusion. (lines 243 -246)

2. The discussion highlights the strong connection between the findings and the Sustainable Development Goals (SDGs).

3. The conclusion of the study can be strengthened by providing explicit and direct recommendations based on its findings. Although advocating for workplace wellness programs is commendable, it would be beneficial to specify precise strategies or areas of focus that can assist in guiding future interventions and policy decisions.

Reviewer 2 ·

Basic reporting

No comments on the content. The Introduction section is well-stated, providing sufficient background information. The English is unambiguous and professional.

Experimental design

Comments: I am impressed by the thorough discussion of essential components for epidemiological research, such as sample size calculation, handling of missing data, and the validity of the survey used to assess the study outcome, as well as the estimation of the confidence interval. The authors have provided sufficient information for reproducibility. However, I do have a few concerns and minor suggestions regarding the method section:

1. Concerning the study design, I am curious about the time point when the outcomes (work productivity and financial loss) were assessed. My major concern is that the study design of the current research does not align perfectly with the definition of a nested case-control study. In this context, a nested case-control study would involve the identification of individuals with incident CVD as cases, the random selection of those without incident CVD as controls, and subsequently tracing back to the baseline to study the effect of covariates assessed during the covariate assessment window on the outcome, incident CVD.
2. I recommend adding headings for the Materials and Methods section, such as data source, outcome, and statistical analysis, to enhance the readability of this section.
3. With regards to the methodology, I am curious about the sample size from each state and how the random selection was conducted. Would a teacher from a state with more teachers in the cohort have a higher or lower probability of being included as a control in the study?

Validity of the findings

no comment

---

## Round 0.2 · Minor Revisions

The authors have addressed all of the reviewers' comments. I have also assessed the revision myself and I am happy with the current version.

However, the Section Editors have noticed an important point which needs your clarification before final acceptance, specifically:

In the Conclusions (both in the abstract and the main manuscript), as well as in the final sentence of the Discussion, the authors advocate for the implementation of strategies targeting the reduction of presenteeism in "cases" (patients suffering from CVD), which already exhibit lower absolute presenteeism compared to "controls" based on the analyses conducted. The authors are encouraged to clarify the rationale and reasons behind this seemingly counterintuitive recommendation, and briefly discuss the potential reasons for the increased cost of presenteeism in "cases", despite the decrease in absolute presenteeism, to enhance understanding for a broader audience

---

## Round 0.3 · accepted · Accept

I confirm that the authors have addressed all of the reviewers' comments. While awaiting the responses from the previous reviewers with respect to the earlier decision of minor revisions, and in the interest of expediting the publication process, I have personally reviewed the reviewers' comments and the authors' responses. In my opinion, the authors have adequately addressed the reviewers' comments, and as such this paper may be accepted for publication.